# Systematic mapping of altermagnetic magnons by resonant inelastic X-ray circular dichroism

Nikolaos Biniskos [1] ✉, Manuel dos Santos Dias [2] ✉, Stefano Agrestini [3], David Sviták [1], Ke-Jin Zhou[3,4], Jiří Pospíšil [1] & Petr Čermák [1]

Altermagnets, a unique class of magnetic materials that combines features of both ferromagnets and antiferromagnets, have garnered attention for their potential in spintronics and magnonics. While the electronic properties of altermagnets have been well studied, characterizing their magnon excitations is essential for fully understanding their behavior and enabling practical device applications. In this work, we introduce a measurement protocol combining resonant inelastic X-ray scattering with circular polarization and azimuthal scanning to probe the chiral nature of the altermagnetic split magnon modes in CrSb. This approach circumvents the challenges posed by domain averaging in macroscopic samples, allowing for precise measurements of the polarization and energy of the magnons in individual antiferromagnetic domains. Our findings demonstrate a pronounced circular dichroism in the magnon peaks, with an azimuthal dependence that is consistent with the theoretical predictions and the *g*-wave symmetry. By establishing a reliable and accessible method for probing altermagnetic magnons, this work opens new avenues for fundamental studies of these collective excitations and for developing next-generation magnonic device applications.

Magnetic materials have long served as a paradigmatic example of broken time-reversal symmetry, traditionally classified as ferromagnets (FMs) or antiferromagnets (AFMs). Recently, however, a distinct class–altermagnets–has emerged, which combine features of both FMs and AFMs, leading to unconventional effects such as the lifting of Kramers spin degeneracy in the electronic band structure[1–6]. Initial studies have predominantly focused on the electronic signatures of altermagnetism–manifest in phenomena such as the anomalous Hall effect[7,8] and X-ray magnetic circular dichroism (XMCD)[9–11]. These effects have been experimentally proven using the isostructural materials MnTe[3,9,12,13] and CrSb[4–6,14]. However, it is equally crucial to develop a direct probe of the fundamental magnetic excitations.

Magnons are the collective excitations of all magnetic materials, and not only underpin many physical properties but should also manifest the symmetry-breaking at the heart of altermagnetism. Their energy spectrum and polarization offer unique insights into the microscopic mechanisms that distinguish altermagnets from conventional AFMs, whose magnonic properties are well understood[15]. The theoretical understanding of altermagnetic magnons is still being developed, with novel behavior due to anisotropies[16], magnon–magnon interactions[17–19], and altermagnetic Stoner excitations[20–22]. Given their central role in enabling low-power, high-speed information processing in magnonic devices and advancing spintronic technologies[23], a detailed investigation of their properties

[1]Department of Condensed Matter Physics, Faculty of Mathematics and Physics, Charles University, Praha, Czechia. [2]Scientific Computing Department, STFC Daresbury Laboratory, Warrington, UK. [3]Diamond Light Source, Harwell Campus, Didcot OX11 ODE, UK. [4]Present address: National Synchrotron Radiation Laboratory and School of Nuclear Science and Technology, University of Science and Technology of China, Hefei, China. ✉e-mail: nikolaos.biniskos@matfyz.cuni.cz; manuel.dos-santos-dias@stfc.ac.uk

is essential for translating altermagnetic phenomena into practical applications.

Current theoretical predictions indicate that the energy splitting of the magnon band structure closely parallels the lifting of Kramers spin degeneracy in the electronic band structure[24]. Detailed simulations have been carried out for materials such as $RuO_2$[25] and MnTe[26,27]. Although the splitting in MnTe was experimentally confirmed[13], the key property of the two magnon branches having opposite circular polarization was not resolved. Distinguishing different polarizations of the magnon branches is challenging by the inevitable presence of multiple altermagnetic domains[28]. To resolve the two split magnon branches predicted by theory, measurements must be performed on a single domain; otherwise, averaging over domains could mask the subtle differences between the modes.

There are two primary techniques for probing magnons: inelastic neutron scattering (INS) and resonant inelastic X-ray scattering (RIXS)[29]. RIXS can probe a wide range of collective excitations, including chiral phonons in quartz[30], spin fluctuations in FeSe and $BaFe_2As_2$[31], and multi-magnon excitations in $\alpha$-$Fe_2O_3$[32,33], with the photon circular polarization helping to distinguish different ferromagnetic magnons in $Fe_3Sn_2$[34]. INS boasts excellent energy resolution, but the neutron beam is typically at least 1 cm in diameter, hence requiring a monodomain sample for magnon polarization studies, which is very challenging to achieve. In contrast, RIXS can be focused on a much smaller spot, allowing for domain-specific measurements. The trade-off, however, is that its energy resolution is poorer, making it difficult to directly resolve small energy gaps such as those observed in MnTe by INS[13].

Here, we introduce a novel and straightforward measurement protocol to demonstrate magnonic altermagnetism. By combining RIXS with circularly polarized X-rays and performing systematic azimuthal scans, we can extract the circular dichroism (CD) of the magnon peaks. Comparing the azimuthal dependence in two different altermagnetic domains allows us to reveal the chiral polarization of the split magnon modes, even if the two peaks are not individually resolvable due to the limited RIXS energy resolution.

CrSb is an ideal candidate for this study, having been experimentally established as an electronic $g$-wave altermagnet[4–6]. Unlike semiconducting MnTe, CrSb is metallic, which results in stronger magnetic exchange interactions. Consequently, its magnons rapidly disperse to higher energies[35], the range ideally suited for investigation with X-rays. This metallic behavior enhances the altermagnetic splitting, making the resulting dichroic signal in RIXS measurements more pronounced and accessible, thereby paving the way for a direct experimental probe of altermagnetic magnon physics.

## Results
### Magnons in CrSb probed by RIXS

CrSb crystallizes in the hexagonal space group $P6_3/mmc$ ($a$ = 4.124(3) Å, $c$ = 5.459(8) Å), with Cr in the Wyckoff position $2a$ and Sb in $2c$ forming a trigonally-distorted octahedron around each Cr[36,37]. Based on neutron diffraction studies, below the Néel temperature ($T_N \approx 690$ K)[38] the magnetically ordered state can be described as ferromagnetic Cr planes which are antiferromagnetically stacked along the $c$-axis[36], and the Cr magnetic moment is 2.7(2) $\mu_B$[36,37]. The crystal

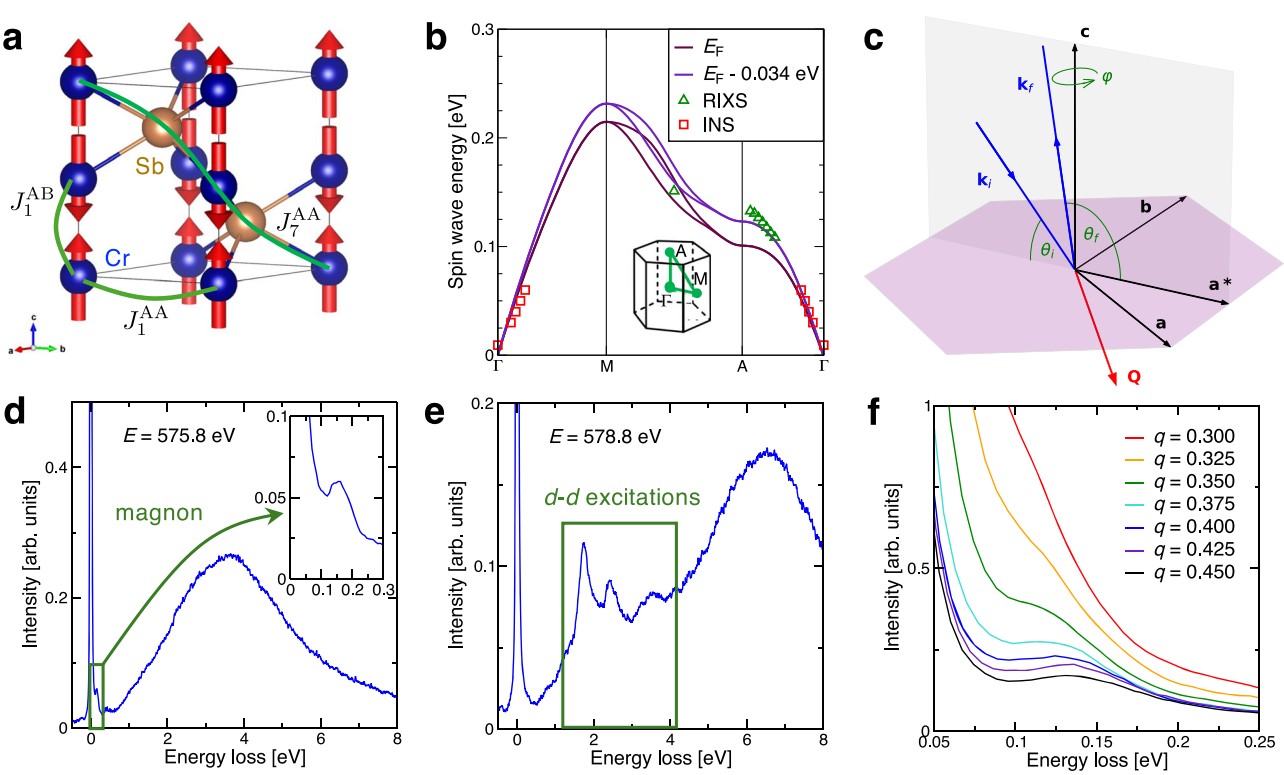

**Fig. 1 | Theoretical and experimental characterization of CrSb. a** Crystal and magnetic structure of CrSb, indicating the nearest-neighbor intra- and inter-sublattice exchange couplings, $J_1^{AA}$ and $J_1^{AB}$, respectively, and the coupling responsible for the altermagnetic magnon splitting, $J_7^{AA}$. **b** Computed magnon bands using the magnetic exchange interactions extracted from the DFT calculations at the theoretical Fermi energy and for a slightly smaller value. Previous INS measurements[35] and our own RIXS measurements are also shown. **c** Scattering geometry for the RIXS experiment. The azimuthal angle $\phi$ is defined to be zero when the scattering plane contains the $a^*$-axis and $\mathbf{Q} \cdot a^* > 0$. **d, e** RIXS spectra showing magnon and $d$–$d$ excitations at two different incident X-ray energies with $\mathbf{Q}$ = (0.27, 0, 0.25) [r.l.u.]. Inset in (**d**) zoom into the low-energy region showing the loss feature due to magnon excitation. **f** Stack plot of representative RIXS spectra showing the evolution of the magnon peaks along the $\Gamma$ - A direction.

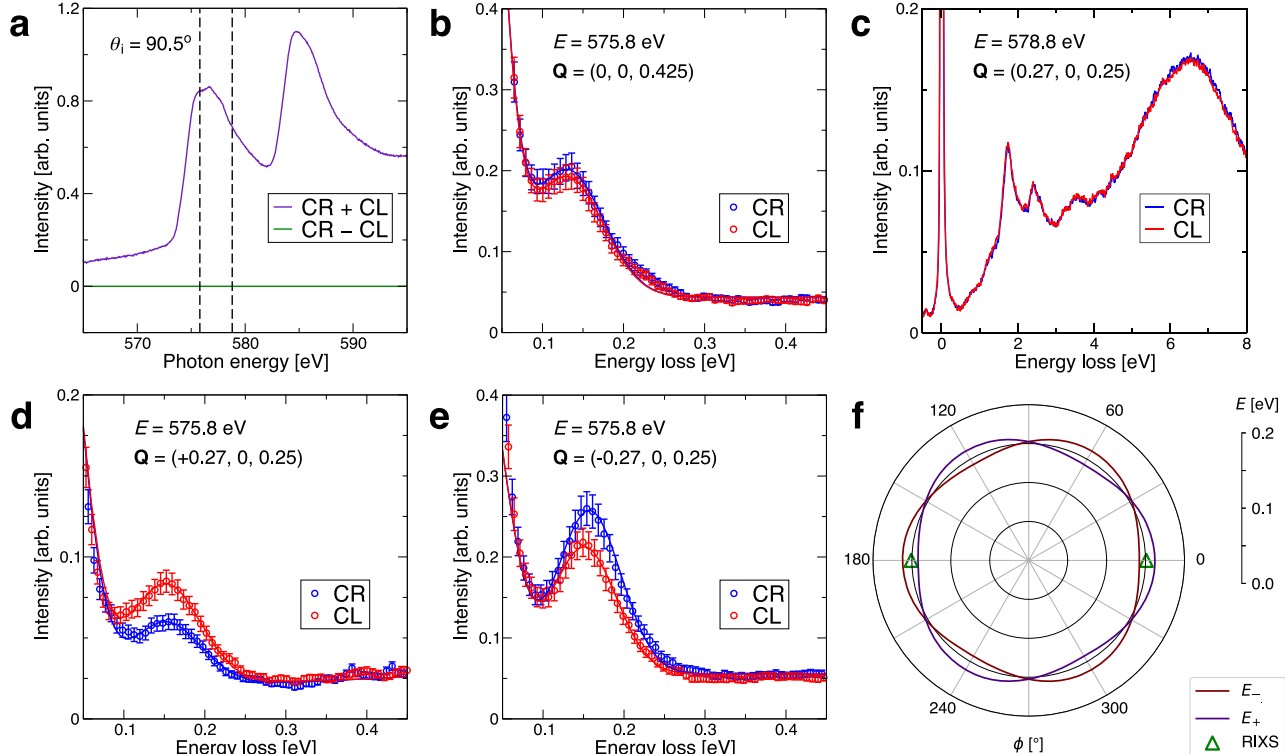

**Fig. 2 | Dichroism in CrSb. a** X-ray absorption and dichroic spectra from total fluorescence yield measurements using incident X-rays with circular-right (CR) and circular left (CL) polarization at normal incidence. The vertical dashed lines indicate the selected photon energies for RIXS. **b** RIXS spectra tuned to a magnon excitation with wave vector along the high-symmetry crystal axis showing no dependence on the circular polarization. **c** RIXS spectra tuned to the $d$–$d$ excitations showing no dependence on the circular polarization. **d, e** RIXS spectra tuned to a magnon excitation with wave vectors with opposite in-plane components show opposite dependence on the circular polarization. **f** Comparison between the theoretical azimuthal dependence of the two altermagnetic magnon bands, $E_{\pm} = E_0 \pm \Delta \cos(3\phi)$, and the magnon excitation energies from the RIXS spectra shown in (**d, e**). In all panels, error bars are standard deviations.

and magnetic structure is depicted in Fig. 1a, together with the most relevant magnetic exchange interactions.

Our DFT calculations reproduce the basic properties of CrSb (See Supplemental material for supporting experimental and theoretical information, which includes refs. 29,39–48). The exchange interactions computed from DFT stabilize the experimental AFM ground state with $T_N \approx 750$ K in the random phase approximation. The respective magnon band structure within linear spin wave theory is shown in Fig. 1b, together with magnon peak positions from previous INS results[35] and our RIXS measurements. The agreement between theory and experiment is good at low energies but worsens for higher energies, in particular near the A point. However, the computed exchange interactions are very sensitive to the precise placement of the Fermi energy. A small downward shift of the Fermi energy leads to an upward shift of the computed magnon energies and better agreement with the RIXS data. Lastly, the magnon degeneracy is present along all high-symmetry lines and lifted elsewhere, with the splitting controlled by the relatively strong $J_7^{AA}$ indicated in Fig. 1a. These theoretical results identify the M−A path as suitable to experimentally investigate altermagnetic magnons.

In order to access magnons at suitably high energy, we performed RIXS measurements on a high-quality CrSb single crystal at the I21 beamline of the Diamond Light Source[39]. The incident photon energy was tuned around the Cr $L_3$ edge, yielding an energy resolution of about 32.5 meV (for more details see Methods and Supplemental material), and we employed both X-ray circular polarizations, circular-right (CR) and circular-left (CL). The scattering geometry is shown in Fig. 1c. Figure 1d, e shows characteristic RIXS spectra for different incident photon energies with $\boldsymbol{Q} = (0.27, 0, 0.25)$ [r.l.u.]. Clear peaks are

observed and identified as magnon (Fig. 1d) and $d$–$d$ (Fig. 1e) excitations. Measurements at different $\boldsymbol{Q} = (0, 0, q)$ positions demonstrate the dispersive nature of the magnons along $\Gamma$ − A, and are presented in Fig. 1f. The resulting magnon peaks were displayed in Fig. 1b and show that the dispersion flattens near the Brillouin zone boundary and reaches an energy of approximately 0.14 eV close to it, in fair agreement with the theoretical dispersions obtained from DFT and connecting to prior INS data[35].

## Circular dichroism in the RIXS spectra

The altermagnetic energy splitting is accompanied by a tendency for each magnon mode to predominantly localize on a given magnetic sublattice and to acquire a well-defined circular polarization or chirality[13] (see also Supplemental material). In a semi-classical picture, this chirality reflects the precession around the exchange field of its dominant sublattice. We can experimentally test this phenomenology by employing X-rays with both circular polarizations, which should couple selectively to the magnon chirality, and by probing the regions of the Brillouin zone where the altermagnetic splitting is predicted to be significant.

We first establish that there is no significant CD otherwise. Figure 2a shows that the conventional XMCD obtained by subtracting the X-ray absorption spectra (XAS) for the two polarizations is indeed negligible, in contrast to the MnTe case[9]. As exemplified in Fig. 2b, we also find no dichroism for the magnon peaks along the $\Gamma$ − A direction, where no altermagnetic character is expected. We next select a momentum transfer $\boldsymbol{Q} = (0.27, 0, 0.25)$ which is inside the $\Gamma$ − M − A plane and for which altermagnetic splittings are allowed, and still find no dichroism for the $d$–$d$ excitations and the broader fluorescence

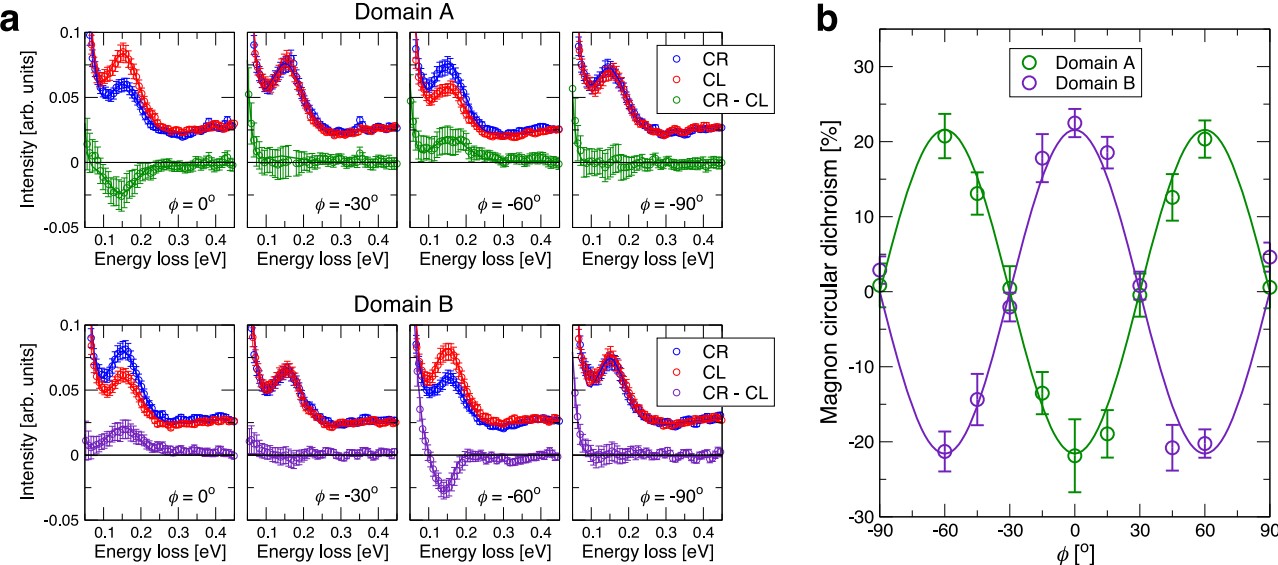

**Fig. 3 | Azimuthal dependence of the RIXS magnon dichroism in CrSb.**
**a** Comparison between RIXS spectra obtained with circularly polarized X-rays in two different AFM domains. Spectra were collected using circular right (CR) and circular left (CL) polarizations, with their intensity difference defining the absolute dichroism, $I_R - I_L$. **b** Azimuthal dependence of the relative circular dichroism, $R_{CD} = 100 \times (I_R - I_L)/(I_R + I_L)$, at the magnon excitation peak for the two opposite antiferromagnetic domains, and fits to the predicted $\cos(3\phi)$ dependence. In all panels, error bars are standard deviations.

peak, as shown in Fig. 2c. However, we do find a pronounced dichroic asymmetry of the magnon peak intensity for the same $\boldsymbol{Q}$, as seen in Fig. 2d, with the CL intensity being about 20% larger than the CR one. Reversing the in-plane component of the momentum transfer, i.e., setting $\boldsymbol{Q} = (-0.27, 0, 0.25)$, leads to a reversal of the dichroic asymmetry, Fig. 2e. As the symmetry is the same, this can be interpreted by comparing with the theoretically determined magnon bands which are plotted in Fig. 2f on a circle of constant $|\boldsymbol{Q}|$, with $\boldsymbol{Q} = (+0.27, 0, 0.25)$ at $\phi = 0°$ and with $\boldsymbol{Q} = (-0.27, 0, 0.25)$ at $\phi = 180°$, with the azimuthal angle $\phi$ being defined in Fig. 1c. Defining $\Delta(\phi) = E_+(\phi) - E_-(\phi)$, the two experimental measurements indeed correspond to a reversal of the chiral nature of the splitting, as $\Delta(180°) = -\Delta(0°)$. The actual peak splitting predicted by the theory of about 22 meV is smaller than the energy resolution of the RIXS experiment (32.5 meV), and so the two independent peaks are not resolvable.

### Systematic mapping of the RIXS magnon dichroism
The inference that the dichroism of the RIXS magnon peak intensity $I_{CD}$ is a proxy for their chiral altermagnetic nature can be further experimentally tested. As symmetry dictates that the magnon chirality has the same angular dependence as the magnon energy splitting, Fig. 2f leads to the predicted azimuthal dependence $I_{CD}(\phi) \propto \cos(3\phi)$, which is the magnon counterpart of the experimentally confirmed electronic $g$-wave azimuthal dependence[4–6]. We thus expect that not only should the dichroism reverse sign for opposite in-plane components of the momentum transfer, but actually reveal a characteristic azimuthal dependence. Furthermore, the dichroism should also reverse if the two sublattices are swapped, that is, if the Néel vector is inverted, as would be found by moving from one antiferromagnetic domain to another.

Using the dichroism of the RIXS magnon peak as a proxy for the orientation of the Néel vector, we performed a scan across the sample and succeeded in locating a different AFM domain (see also Supplemental material), which we term domain B, with the measurements described so far ascribed to domain A. We anticipated that large magnetic domains should exist in CrSb, due to its similar magnetic properties to $Cr_2O_3$ for which they have been imaged[40]. Representative RIXS spectra on both domains for CR and CL polarization of the incident beam are shown in Fig. 3a. The dichroic signal of domain B has the

opposite sign to that of domain A, and vanishes below the error bar for the high-symmetry directions $\phi = -30°$ and $-90°$, as predicted. The extracted relative CD, $R_{CD} = 100 \times (I_R - I_L)/(I_R + I_L)$, is plotted in Fig. 3b. The experimental azimuthal dependence can be perfectly reproduced by $R_{CD} = A\cos(3\phi)$, where $|A| = (21.6 \pm 1.7)\%$.

## Discussion
These observations demonstrate unambiguously that RIXS with circularly polarized X-rays can be used to probe the chirality switching of the altermagnetic magnons. Although RIXS is a complex photon-in/photon-out process which is difficult to interpret quantitatively[29], according to Haverkort[41] it essentially probes the dynamical structure factor, and hence the magnon spectral density, in the energy loss range of interest. This theory explains the RIXS process in terms of low-energy effective scattering operators $\mathcal{R}_\mu = \sigma_\mu^0 (\boldsymbol{\epsilon}_f^* \cdot \boldsymbol{\epsilon}_i) + \sigma_\mu^{CD}(\boldsymbol{\epsilon}_f^* \times \boldsymbol{\epsilon}_i) \cdot \boldsymbol{S}_\mu + \mathcal{O}(S^2)$, where $\sigma_\mu^0$ and $\sigma_\mu^{CD}$ are the fundamental isotropic and dichroic X-ray spectral functions for each sublattice ($\mu = $ A, B), $\boldsymbol{\epsilon}_i$ and $\boldsymbol{\epsilon}_f$ are the polarization vectors for the incident and outgoing photons, respectively, and $\boldsymbol{S}_\mu$ is the spin operator. The atomic spin thus couples to the fundamental X-ray circular dichroic spectrum, establishing a link to magnetic CD.

What is unique to altermagnets is the existence of a mechanism for breaking the balance between the two contributions arising from each sublattice. For a conventional collinear AFM, the dichroic contribution of sublattice B is equal in magnitude and opposite in sign to that of sublattice A, and so there is no net dichroic signal. Nevertheless, a small XMCD signal has been found in the conventional X-ray absorption spectrum of altermagnetic MnTe, but it was explained to arise from either core–valence interactions or the staggered Weiss field[9]. For the altermagnetic candidate $MnF_2$ XMCD has been theoretically predicted[10] and measured a long time ago but with visible light[49], while no splitting was observed in the magnon spectrum[50].

We instead propose a different and much stronger mechanism to explain our substantial CD measured at the RIXS magnon peaks. There are two ways in which altermagnetism can lead to asymmetric behavior in the magnon spectrum: by lifting the magnon energy degeneracy[25] and by leading to mode-dependent linewidths which can become quite different if the magnons interact with the altermagnetic Stoner continuum[20–22]. These mechanisms are absent in conventional collinear AFMs, and both result in a finite CD of the RIXS magnon peak, with

the experimental signal favoring the explanation based on asymmetric linewidths. We present a self-contained derivation of this result in the Supplemental material, a simple expression for $R_{CD}(q, \omega)$ in Eq. S53, and a toy model illustration of the two scenarios in Fig. S4. This explains the correlation between the azimuthal dependence of the experimental CD (Fig. 3b) and the azimuthal dependence of the two magnon bands and their energy splitting (Fig. 2f). The much larger magnon energies in CrSb in comparison to the ones measured by INS MnTe[13] are a key aspect enabling the detection of the magnon dichroism in RIXS, given the experimental energy resolution. The other favorable aspect is the simpler domain structure in CrSb (there are only two domains with the Néel vector along the c-axis), while MnTe has six in-plane domains[28] which complicate the interpretation of its dichroism[51,52].

In this work, we established a connection between the CD of the magnon peak detected in RIXS and the predicted altermagnetic properties of the two circularly-polarized magnons in the g-wave altermagnet CrSb. This material offers a versatile platform to explore the interplay between various effects enabled by altermagnetism, such as Stoner damping[20–22], its properties being tunable by chemical substitution[53] or as part of heterostructures[14]. Our experimental approach of mapping the azimuthal dependence of the dichroic signal avoids potential spurious contributions from birefringence in RIXS[54], which is independent of the azimuthal angle in the chosen scattering configuration. This defines a robust protocol for the investigation of dichroic properties using RIXS in other altermagnetic material candidates. To conclude, our discovery of CD at the magnon peak completes the experimental demonstration that altermagnetic magnons indeed possess all their predicted properties—energy splitting (proven in ref. 13) and circular polarization (present study)—, opening exciting perspectives for the development of altermagnetic magnonics[55] and its combination with spintronics[56].

## Methods

### Sample preparation and experimental details

Single crystals of CrSb were grown by the chemical vapor transport technique using iodine as a transporting agent. A CrSb crystal was aligned in the [100]/[001] scattering plane using X-ray Laue diffraction at room temperature prior to the RIXS measurements. In this article, we use the hexagonal coordinate system, and the scattering vector $Q$ is expressed in $Q = (Q_h, Q_k, Q_l)$ given in reciprocal lattice units (r.l.u.). RIXS was performed at the I21 beamline of the Diamond Light Source in the UK[39]. The used photon energy was around the Cr $L_3$ edge, and polarization was circular (CR/CL). The energy resolution was estimated as 32.5 meV from the full-width of the half-maximum of the elastic peak from a carbon tape. XAS was performed prior to the RIXS measurements and is based on the total fluorescence yield method. All data were obtained at $T = 300$ K and at zero magnetic field. Further details can be found in the Supplemental material.

### Theoretical details

The theoretical results were obtained with DFT calculations and the extracted spin Hamiltonian. The ground state properties and the magnetic exchange interaction parameters were computed with the DFT full-potential all-electron code JuKKR[42,43,57], employing the experimental crystal structure parameters. The exchange interactions are then used to solve a spin Hamiltonian in the linear spin wave approximation[58]. Further details can be found in the Supplemental material.

## Data availability

The authors declare that the data supporting the findings of this study are available within the paper, its supplementary information file which contains additionally references[44–48] and in the following repository[60].

## Code availability

The DFT simulation package juKKR is publicly available (see "Methods"). The code for the solution of the linear spin wave problem is available from the corresponding authors upon request.

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

## Acknowledgements

We thank Dr. Mirian García-Fernández for discussions and comments during the beam time. We acknowledge Diamond Light Source for providing the beam-time at beamline I21 under Proposal MM38040-1. This work was supported by the Czech Science Foundation GAČR under the Junior Star Grant No. 21-24965M (MaMBA). Crystals were grown and characterized in MGML, which is supported within the program of Czech Research Infrastructures (project No. LM2023065). The work of M.d.S.D. made use of computational support by CoSeC, the Computational Science Centre for Research Communities, through CCP9 (EPSRC EP/T026375/1)[59]. Computing resources were provided by STFC Scientific Computing Department's SCARF cluster. The work of D.S. and P.Č. was supported by the bilateral Czech-Bavarian project BaCQuERel (project No. LUABA24056).

## Author contributions

N.B. and M.d.S.D. contributed equally to this work. N.B., M.d.S.D. and S.A. conceived the project together with K.-J.Z. and P.Č. N.B., M.d.S.D., S.A. and D.S. performed the experimental measurements and N.B. analyzed the corresponding data. S.A. obtained the XMCD signal by subtracting the relevant XAS. M.d.S.D. performed the DFT calculations and the spin-wave modeling. S.A. and K.-J.Z. were local contacts of the I21 beam-line and provided instrument support. J.P. grew the single crystals and J.P., N.B. and D.S. characterized them with in-house methods. P.Č. supervised the project. All authors participated in the discussion of the results. M.d.S.D. and N.B. wrote the manuscript with input from all authors.

## Competing interests

The authors declare no competing interests.
