## [Transparent Peer Review file · Nature Communications]

Systematic Mapping of Altermagnetic Magnons by Resonant Inelastic X-Ray Circular Dichroism

Corresponding Author: Dr Nikolaos Biniskos

Version 0:

Reviewer comments:

Reviewer #1

(Remarks to the Author)

The authors present RIXS measurements on magnons in the altermagnet CrSb. They report that RIXS-MCD signals reverse sign at $\pm Q$ and exhibit a clear azimuthal dependence, which they interpret as a direct observation of altermagnetic magnon chirality. The experimental data are well-presented, and the manuscript is clearly structured. The central claim regarding the experimental observation of RIXS-MCD in magnon is evident.

While the focus on experimental validation without comprehensive theoretical detail is acceptable at this stage, several questions arise concerning the qualitative interpretation of the results, particularly given the insufficient energy resolution. Therefore, I recommend acceptance only after the authors have adequately addressed the following points:

- 1) Interpretation of RIXS-MCD from Unresolved Peaks: A fundamental question concerns the interpretation of RIXS-MCD arising from unresolved magnon peaks. Typically, MCD reflects the intensity difference for a given feature measured with right (CR) and left (CL) circularly polarized light. Here, the authors state that the predicted altermagnetic magnon modes (E+ and E-) are split but cannot be resolved due to the insufficient energy resolution. Thus, the observed magnon feature in Figs. 2 and 3 presumably contains contributions from both E+ and E-. Given that these modes are expected to have opposite chirality, one might naively expect their individual MCD contributions to be opposite in sign. If so, how does a net, non-zero MCD signal arise when integrating over the unresolved peak? Could the authors clarify the relationship between the unresolved nature of the E+ and E- modes and the observation of a significant net R_CD (quantified as 21.6%)? How should this value be interpreted as potentially representing the sum of two opposing contributions?
- 2) Apparent Peak Shift in MCD Spectra: Related to point (1), when looking closely at Figs. 2d and 2e (particularly 2e), there appears to be a slight shift in the peak maximum position between the CR and CL spectra for the magnon feature. Is this clear shift statistically significant? Could such a shift arise naturally if the unresolved E+ and E- peaks have substantially different intensities under CR and CL excitation (due to the MCD effect), thereby shifting the center-of-mass of the combined observed peak?
- 3) Intensity Normalization: Could the authors please clarify the procedure used for intensity normalization of the RIXS spectra? (e.g., normalization to incident flux, acquisition time, elastic peak, fluorescence features?)
- 4) Positional Stability during Azimuthal Scans: The azimuthal scans are critical to the analysis. What measures were taken to ensure that the X-ray beam remained on the exact same sample spot (and thus within the same magnetic domain, e.g., Domain A or B) throughout the azimuthal rotation? How was the positional stability confirmed during these scans?

Reviewer #2

(Remarks to the Author)

Please see the attachment.

Reviewer #3

(Remarks to the Author)

The manuscript employs a novel experimental approach to confirm theoretically predicted properties of magnons in altermagnetic materials. Such magnons are very recently opened direction of research with the "hot topic" of altermagnets where unusual their unusual properties, such as energy splitting and circular polarization, require particular scrutiny. The proposed experimental technique makes it possible to bypass problems which standard neutron scattering encounters, particularly the issue of domain averaging in macroscopic samples, which can obscure detecting special features of altermagnetic magnons.

The manuscript is carefully constructed, with comparison between theory and measurements, and sufficient details provided so that other groups can employ the same technique in future probing of altermagnetic magnons. This makes it publishable. However, some improvements are needed.

The authors assume that linear spin wave theory (LSWT), as computed in Fig. 1b, is sufficient to describe altermagnetic magnons. This could have also been motivated by early theoretical studies using the same level of description as in Ref. 18. However, recent scrutiny shows that altermagnetic magnons can exhibit highly nontrivial many-body effects (i.e., beyond LSWT). This is astounding, as such effects typically require noncollinearity and/or spin-orbit coupling. Thus, this topic has ignited a recent flurry of theoretical papers, such as:

1. Magnon spectrum of altermagnets beyond linear spin wave theory: Magnon-magnon interactions via time-dependent matrix product states versus atomistic spin dynamics, Federico Garcia-Gaitan, Ali Kefayati, John Q. Xiao, and Branislav K. Nikolic, Phys. Rev. B 111, L020407 – Published 15 January, 2025.

2. Spontaneous Magnon Decays from Nonrelativistic Time-Reversal Symmetry Breaking in Altermagnets, Rintaro Eto, Matthias Gohlke, Jairo Sinova, Masahito Mochizuki, Alexander L. Chernyshev, Alexander Mook; <https://arxiv.org/abs/2502.20146>

3. Spontaneous magnon decay in two-dimensional altermagnets
Niklas Cichutek, Peter Kopietz, Andreas Rückriegel; <https://arxiv.org/abs/2502.19815>

None of them are mentioned in the manuscript, but could be very relevant, such as:

a) with neutron scattering, finite magnon lifetime due to spontaneous decay has been measured in many antiferromagnets. Can this new experimental approach shed light on the same (inevitable, according to 1.-3. above) property of altermagnetic magnons.

b) Some discrepancy between LSWT and experimental data is already found in Fig. 1b of the manuscript.

Regarding formatting, the manuscript lists some references with full names of authors (such as [18] Libor Š 296 mejkal, Alberto Marmodoro, Kyo-Hoon Ahn ...), which is non-standard practice, then reverts to standard practice in other cases (such as [50] K. Wildberger, P. Lang, R. Zeller, and P. H. Dederichs, ...).

Version 1:

Reviewer comments:

Reviewer #1

(Remarks to the Author)

The authors have diligently addressed all comments from the reviewers and have revised the manuscript accordingly. While the proposed explanation—that the MCD arises from a difference in magnon lifetimes—is plausible and self-consistent, it is, as Reviewer #2 also pointed out, a rather unconventional interpretation. My primary concern with the theoretical interpretation—specifically its lack of quantitative support—remains unchanged. The quality of the experimental data itself is not in question. The issue lies with the conclusive tone used to present the interpretation, which I believe is too strong given the nature of the evidence.

This interpretation is presented too conclusively in several key places. Specifically, I find the following phrases to be overstated:

In the Conclusion, the statement that your work "...completes the experimental demonstration...".

In the Introduction and Discussion, the use of "...unambiguously demonstrate...".

In the Discussion, the claim to have "...demonstrated a direct connection...".

In the Abstract, the phrase "...directly confirms the theoretical predictions...".

The claim of having "completes the experimental demonstration" is particularly premature. A 'complete' demonstration would arguably require resolving the split peaks themselves, which should be achievable with an optimal combination of sample quality and a state-of-the-art RIXS instrument capable of sub-20 meV resolution at the Cr L-edge. Without such direct observation, the other strong phrases like "unambiguously demonstrate" and "direct connection" are similarly not yet fully justified.

Therefore, I would be prepared to recommend acceptance of this manuscript on the condition that these and similar interpretive claims are softened. The focus should be placed on the high-quality experimental observation of MCD, which is an excellent result in itself, while presenting the origin as a plausible but not yet definitive explanation.

Replacing these strong claims with more cautious phrasing—such as "our results suggest...", "we propose a scenario where...", or "the data is consistent with..."—would make the manuscript more robust and accurately reflect the current state of evidence. I hope these specific comments are helpful for further improving this important work.

Reviewer #2

(Remarks to the Author)

The authors proposed a good idea that bypasses the problem raised by the reviewer #1 and me. I'm very pleased to recommend publication of the revised article in Nature Communications.

Reviewer #3

(Remarks to the Author)

The authors have resolved all technical questions of 3 referees, and correspondingly improved the manuscript. There is no doubt that their effort to introduce novel techniques for probing chiral magnons (an emerging topic on its own, of potential application significance as well for magnonic technologies) in altermagnets could have a large and broad impact. This has been recognized already, as in the viewpoint, <https://www.condmatclub.org/2025/06/>, written for the highly selective Journal Club for Condensed Matter Physics. So, I recommend the paper be published as is.

Version 2:

Reviewer comments:

Reviewer #1

(Remarks to the Author)

Reviewer #2

(Remarks to the Author)

I perfectly agree with the comments raised by the Reviewer #1. The revised manuscript is surely improved compared with the previous one. I recommend publication of this article in Nature Communication.

Response to the reviewers

We thank the reviewers for their constructive feedback. Below, we address each comment point by point. The reviewers' reports are reproduced in black, our responses are in blue, and all changes to the manuscript and the Supplementary Information are highlighted in red. The reference numbers correspond to the revised manuscript.

Report of Reviewer #1

The authors present RIXS measurements on magnons in the altermagnet CrSb. They report that RIXS-MCD signals reverse sign at $\pm Q$ and exhibit a clear azimuthal dependence, which they interpret as a direct observation of altermagnetic magnon chirality. The experimental data are well-presented, and the manuscript is clearly structured. The central claim regarding the experimental observation of RIXS-MCD in magnon is evident.

While the focus on experimental validation without comprehensive theoretical detail is acceptable at this stage, several questions arise concerning the qualitative interpretation of the results, particularly given the insufficient energy resolution. Therefore, I recommend acceptance only after the authors have adequately addressed the following points:

We thank the reviewer for recognizing the relevance of our experimental results and for pointing out aspects of the theoretical interpretation that deserve further comment. In the following we address all these points in detail.

1) Interpretation of RIXS-MCD from Unresolved Peaks: A fundamental question concerns the interpretation of RIXS-MCD arising from unresolved magnon peaks. Typically, MCD reflects the intensity difference for a given feature measured with right (CR) and left (CL) circularly polarized light. Here, the authors state that the predicted altermagnetic magnon modes (E+ and E-) are split but cannot be resolved due to the insufficient energy resolution. Thus, the observed magnon feature in Figs. 2 and 3 presumably contains contributions from both E+ and E-. Given that these modes are expected to have opposite chirality, one might naively expect their individual MCD contributions to be opposite in sign. If so, how does a net, non-zero MCD signal arise when integrating over the unresolved peak? Could the authors clarify the relationship between the unresolved nature of the E+ and E- modes and the observation of a significant net R_{CD} (quantified as 21.6%)? How should this value be interpreted as potentially representing the sum of two opposing contributions?

This is a very pertinent question that we strove to explain in the Discussion part in the first version of the manuscript: "As in conventional collinear AFMs [37], the altermagnetic magnon modes predominantly localize on a given sublattice. However, due to the altermagnetic lifting of the magnon energy degeneracy the two magnon modes contribute to the spectrum in an unequal way, and in combination with their selective sublattice localization results in a finite circular dichroism of the RIXS magnon peak. We present a self-contained derivation of this result in the Supplementary Information [39], and a simple expression for $R_{CD}(\mathbf{q}, \omega)$ in Eq. S53. This explains the correlation between the azimuthal dependence of the experimental circular dichroism (Fig. 3(b)) and the azimuthal dependence of the two magnon bands and their energy splitting (Fig. 2(f))."

We now realise that this level of detail is insufficient for a reader to follow the argument without consulting the Supplement. Our explanation for the circular dichroism relies on: (i) the

altermagnetic magnons are not degenerate in either energy or linewidth, except where this is symmetry enforced; (ii) the altermagnetic magnons are intrinsically chiral, in the sense that a definite amount of spin angular momentum can be assigned to each of them, $\Delta M_S = \pm 1$ (see also Kravchuk et al., arXiv:2504.05241; Ref. [16]). Qualitatively, this spin angular momentum is proportional to the circular dichroism and so detectable via RIXS at the respective magnon peaks, as long as they are not identical. If the splitting between the two magnon peaks were larger than the combination of their intrinsic linewidth and the experimental energy resolution one would expect a pronounced reversal in the sign of the MCD between the two peaks, as also highlighted by Reviewer #2. Conversely, if there were no energy splitting between the peaks the net MCD would cancel, assuming that the linewidths are the same, and this is what happens for high-symmetry scattering configurations such as $\phi = \pm 30^\circ, \pm 90^\circ$, as observed. We thus suppose that the energy splitting is finite but below the experimental energy resolution and also smaller than the theoretically computed value. This points to the dichroism coming from different linewidths instead, which is also covered by our Eq. S53 with appropriate parameters. This has indeed been predicted to happen for metallic altermagnets (Costa et al., SciPost Phys. **18**, 125 (2025); this is Ref. [20]). Two recent preprints report detailed calculations for the impact of Stoner excitations on the altermagnetic magnons of CrSb and align with our proposed explanation (Zhang et al., Phys. Rev. B **111**, 174451 (2025), Ref. [21], and Beida et al., arXiv:2505.08103, Ref. [22]). We provide an illustration of the two scenarios in Fig. R1.

In order to clarify this point, we revised this discussion in the manuscript and added Fig. R1 to the supplement as Fig. S4:

“We instead propose a different and much stronger mechanism to explain our substantial circular dichroism measured at the RIXS magnon peaks. There are two ways in which altermagnetism can lead to asymmetric behaviour in the magnon spectrum: by lifting the magnon energy degeneracy [25] and by leading to mode-dependent linewidths which can become quite different if the magnons interact with the altermagnetic Stoner continuum [20–22]. These mechanisms are absent in conventional collinear AFMs, and both result in a finite circular dichroism of the RIXS magnon peak, with the experimental signal favoring the explanation based on asymmetric linewidths. We present a self-contained derivation of this result in the Supplementary Information [39], a simple expression for $R_{CD}(\mathbf{q}, \omega)$ in Eq. S53, and a toy model illustration of the two scenarios in Fig. S4.”

2) Apparent Peak Shift in MCD Spectra: Related to point (1), when looking closely at Figs. 2d and 2e (particularly 2e), there appears to be a slight shift in the peak maximum position between the CR and CL spectra for the magnon feature. Is this clear shift statistically significant? Could such a shift arise naturally if the unresolved E+ and E- peaks have substantially different intensities under CR and CL excitation (due to the MCD effect), thereby shifting the center-of-mass of the combined observed peak?

We thank the reviewer for this careful observation. We have analysed the data as a combination of various features: elastic line, magnon peak, fluorescence peak. The magnon peak is not very intense, and so its peak parameters (position, linewidth, intensity) are quite sensitive on how the other two contributions are fitted and define the respective background upon which the magnon peak rests. We thus concluded that the small apparent magnon peak shifts could not be given significance in a reliable way. As the reviewer remarks, it is quite possible that the apparent position of the magnon peak would shift slightly due to the different intensities of the two underlying peaks, but this could not be made quantitative.

3) Intensity Normalization: Could the authors please clarify the procedure used for intensity

Fig. R1: Two possible scenarios for circular dichroism in the RIXS spectra. (a) Peaks centered at different energies but with same broadening: $\omega_0 = 0.14, 0.16$ eV, $\Gamma = 0.033$ eV, equivalent to Fig. R2 by Reviewer #2. (b) Peaks centered at same energy but with different broadenings: $\omega_0 = 0.15$ eV, $\Gamma = 0.03, 0.04$ eV. The peaks are given by $I_{\text{peak}}(\omega) = \frac{\omega/\pi}{(\omega-\omega_0)^2+\Gamma^2}$. In both scenarios a simple background representing the tail of the elastic line was included, with the form $I_{\text{bg}}(\omega) = \frac{0.5}{\omega^2}$. The peak average and the relative difference, $100 \times (\text{peak1} - \text{peak2}) / (\text{peak1} + \text{peak2})$, which mimics R_{CD} defined from the RIXS spectra (see Eq. S53 in the supplement), are also shown.

normalization of the RIXS spectra? (e.g., normalization to incident flux, acquisition time, elastic peak, fluorescence features?)

The information regarding the intensity normalization was missing. We added in the supplement the following:

“Every RIXS spectrum is normalised to the incident flux, measured via the focusing mirror current, which was collected simultaneously with the RIXS spectrum with the same acquisition time. In short, the RIXS intensity is divided by the M4 mirror current.”

4) Positional Stability during Azimuthal Scans: The azimuthal scans are critical to the analysis. What measures were taken to ensure that the X-ray beam remained on the exact same sample spot (and thus within the same magnetic domain, e.g., Domain A or B) throughout the azimuthal rotation? How was the positional stability confirmed during these scans?

The information regarding the positional stability during the azimuthal scans was missing. We added in the supplement the following:

“Using the dichroism of the RIXS magnon peak as a proxy for the orientation of the Néel vector, a scan across the sample allowed us to determine that the domain sizes in CrSb are of the order of several hundreds of micrometres, i.e. much larger than the small footprint of the beam. For the azimuthal scans two positions of the domains A and B were selected well away from the domain border and separated by $500 \mu\text{m}$ (see Fig. S1). A cross overlap to the image of the optical micro-camera provides a visual view of the location of the beam on the sample. The beam position on the sample was recorded and checked using the optical camera after

each azimuthal rotation. There was a small drift in the sample position after each rotation but given the large domain separation and the small beam size, all the RIXS MCD were confidently measured well within each single magnetic domain (A or B).”

Report of Reviewer #2

In this paper, Biniskos et al. carry out careful resonant inelastic x-ray scattering measurements and report unambiguous circular dichroism in magnon intensity in the altermagnet CrSb by particularly performing wide azimuthal scans for two opposite altermagnetic domains. This subject is very topical in condensed matter physics and hence is intensively studied. Although, as will be mentioned, I think that the theoretical treatment is insufficient, the dichroic features are exhibited in magnon intensity without ambiguity. From an experimental point of view, this is an important progress, which merits publication in Nature Communications. I think that several revisions are necessary. Please carefully consider following comments and correctly reflect them in the revised manuscript.

We thank the reviewer for the positive assessment of our experimental findings and for bringing to our attention several aspects that need clarification. In the following we carefully address all the raised points.

(1) The authors should distinguish “splitting” from “dichroism in intensity”.

We agree with the reviewer’s remark that we should have been more careful in separating the experimentally observed “dichroism in intensity” from the theoretically predicted “energy splitting”. Both aspects arise from the same fundamental origin, which is altermagnetism, and have the same symmetry properties. Our derivation of the altermagnetic magnon properties provides the fundamental link between the lifting of the energy degeneracy and the magnon chirality, hence we can use one property as a proxy for the other. We next comment on the manuscript text highlighted by the reviewer:

line 116-117: This can be interpreted by comparing with the theoretically determined magnon bands

This is correct in the sense explained in our previous reply, that one property tracks the other and they share the same symmetry properties, hence the dichroism can be interpreted to arise from altermagnetic magnons and is expected to arise whenever the energy degeneracy is allowed to be lifted by symmetry. We made the following modification:

“As the symmetry is the same, this can be interpreted by comparing with the theoretically determined magnon bands [...]”

line 126-127: Fig. 2(f) leads to the predicted azimuthal dependence $I_{CD}(\phi) \propto \cos(3\phi)$, Fig. 2(f) indicates the azimuthal dependence of energy splitting in magnon bands. Hence it does not (simply) explain the observed dichroism in intensity and its ϕ dependence.

The relationship is indirect, as our explanation is that the dichroism is related to the magnon chirality which in turn is relevant whenever the magnon energy degeneracy is lifted. To clarify, we have modified this line as follows:

“By symmetry, the magnon chirality has the same angular dependence as the magnon energy splitting, Fig. 2(f) leads to the predicted azimuthal dependence $I_{CD}(\phi) \propto \cos(3\phi)$, which is the magnon analogue of the experimentally confirmed electronic g-wave azimuthal dependence

[4–6] [...]”

line 190-191: altermagnetic magnons indeed possess all their predicted properties – energy splitting and circular polarization –,

By this we meant that our experimental detection of the altermagnetic magnon chirality via the circular dichroism of the RIXS spectra complement the inelastic neutron scattering measurement of the altermagnetic magnon energy splitting for MnTe (Liu et al., Phys. Rev. Lett. **133**, 156702 (2024); Ref. [13]). We modified that statement as follows:

“To conclude, our discovery of circular dichroism at the magnon peak completes the experimental demonstration that altermagnetic magnons indeed possess all their predicted properties — energy splitting (proven in Ref. [13]) and circular polarization (present study) [...]”

The authors observed the circular dichroism in magnon intensity, but the energy splitting in magnon bands was not detected in this paper.

The reviewer is correct and we have made our statements more precise as explained above.

Fig. R2: [From Reviewer #2.] Expected dichroic spectrum (dark green) for split peaks with opposite chirality (red and blue). Lorentzians with $\Gamma = 33$ meV. Energy splitting is 22 meV.

(2) If my understanding is correct, $R_{CD}(\omega)$ in Eq. S53 should be a dark green line indicated in Fig. R2. However, instead of this negative-positive (or positive-negative) spectrum, a single peak is observed in dichroic intensity. Therefore, the authors should describe in the main text that the theoretical calculation presented in the Supplementary Information does not interpret the experimental observation and more sophisticated theoretical approach is necessary. Accordingly,

line 172-174: This explains the correlation between the azimuthal dependence of the experimental circular dichroism (Fig. 3(b)) and the azimuthal dependence of the two magnon bands and their energy splitting (Fig. 2(f)).

This is not correct.

Abstract line 11: the theoretical predictions and

This is not correct.

The reviewer raises a very pertinent point. There are actually two different altermagnetic mechanisms by which the observed dichroism can arise. The magnon energies can be different

(lifting of the energy degeneracy) but the magnon linewidths can also be different (from the broadening caused by Stoner excitations, which are themselves altermagnetic. See Costa et al., *SciPost Phys.* **18**, 125 (2025); this is Ref. [20]). The two limiting scenarios are illustrated in Fig. R1. The energy splitting scenario is the one highlighted by the reviewer, and reproduced as Fig. R1(a). This scenario does indeed lead to a negative-positive sign change in R_{CD} . The linewidth splitting scenario is shown in Fig. R1(b) and leads to a positive R_{CD} .

The experimental data resembles scenario (b). This suggests that the actual magnon energy splitting might be smaller than the theoretically predicted value of about 0.02 eV, which is in any case smaller than the experimental energy resolution of 0.03 eV. There is also a theoretical explanation for this: according to the DFT simulations, the energy splitting becomes very small as soon as the Fermi energy shifts to below the lower edge of the minority Cr d -states, which could happen due to a small deviation from perfect stoichiometry. While this shift in Fermi energy would suppress the magnon energy splitting, it would not have the same impact on the Stoner excitations, which should remain robustly altermagnetic. Although we did not perform our own calculation of the effect of the Stoner excitations, two other groups have uploaded preprints that report this type of calculation (Zhang et al., *Phys. Rev. B* **111**, 174451 (2025), Ref. [21], and Beida et al., arXiv:2505.08103, Ref. [22]), and both predict large and quite different linewidths for each magnon band.

In order to clarify this point, we added the following discussion to the manuscript and Fig. R1 to the supplement:

“We instead propose a different and much stronger mechanism to explain our substantial circular dichroism measured at the RIXS magnon peaks. There are two ways in which altermagnetism can lead to asymmetric behaviour in the magnon spectrum: by lifting the magnon energy degeneracy [25] and by leading to mode-dependent linewidths which can become quite different if the magnons interact with the altermagnetic Stoner continuum [20–22]. These mechanisms are absent in conventional collinear AFMs, and both result in a finite circular dichroism of the RIXS magnon peak, with the experimental signal favoring the explanation based on asymmetric linewidths. We present a self-contained derivation of this result in the Supplementary Information [39], a simple expression for $R_{CD}(\mathbf{q}, \omega)$ in Eq. S53, and a toy model illustration of the two scenarios in Fig. S4.”

Report of Reviewer #3

The manuscript employs a novel experimental approach to confirm theoretically predicted properties of magnons in altermagnetic materials. Such magnons are very recently opened direction of research with the “hot topic” of altermagnets where unusual their unusual properties, such as energy splitting and circular polarization, require particular scrutiny. The proposed experimental technique makes it possible to bypass problems which standard neutron scattering encounters, particularly the issue of domain averaging in macroscopic samples, which can obscure detecting special features of altermagnetic magnons.

The manuscript is carefully constructed, with comparison between theory and measurements, and sufficient details provided so that other groups can employ the same technique in future probing of altermagnetic magnons. This makes it publishable. However, some improvements are needed.

We thank the reviewer for the positive assessment of our work and for bringing to our attention the potential role of effects beyond linear spin wave theory. In the following we reply to this issue in detail.

The authors assume that linear spin wave theory (LSWT), as computed in Fig. 1b, is sufficient to describe altermagnetic magnons. This could have also been motivated by early theoretical studies using the same level of description as in Ref. 18. However, recent scrutiny shows that altermagnetic magnons can exhibit highly nontrivial many-body effects (i.e., beyond LSWT). This is astounding, as such effects typically require noncollinearity and/or spin-orbit coupling. Thus, this topic has ignited a recent flurry of theoretical papers, such as:

1. Magnon spectrum of altermagnets beyond linear spin wave theory: Magnon-magnon interactions via time-dependent matrix product states versus atomistic spin dynamics, Federico Garcia-Gaitan, Ali Kefayati, John Q. Xiao, and Branislav K. Nikolic, Phys. Rev. B 111, L020407 – Published 15 January, 2025.
2. Spontaneous Magnon Decays from Nonrelativistic Time-Reversal Symmetry Breaking in Altermagnets, Rintaro Eto, Matthias Gohlke, Jairo Sinova, Masahito Mochizuki, Alexander L. Chernyshev, Alexander Mook; <https://arxiv.org/abs/2502.20146>
3. Spontaneous magnon decay in two-dimensional altermagnets Niklas Cichutek, Peter Kopietz, Andreas Rückriegel; <https://arxiv.org/abs/2502.19815>

None of them are mentioned in the manuscript, but could be very relevant, such as:

- a) with neutron scattering, finite magnon lifetime due to spontaneous decay has been measured in many antiferromagnets. Can this new experimental approach shed light on the same (inevitable, according to 1.-3. above) property of altermagnetic magnons.
- b) Some discrepancy between LSWT and experimental data is already found in Fig. 1b of the manuscript.

We agree that LSWT can be insufficient for altermagnets, as convincingly argued in the works cited by the reviewer. However, the situation is likely even more complex. Besides the potential beyond-LSWT effects, Stoner excitations can also have a strong impact which is only now beginning to be explored (see Costa et al., SciPost Phys. **18**, 125 (2025); this is Ref. [20]. See also the recent works by Zhang et al., Phys. Rev. B **111**, 174451 (2025), Ref. [21], and Beida et al., arXiv:2505.08103), Ref. [22]. The interaction between magnons and Stoner excitations can also lead to both strong damping and renormalization of the excitation energies, and so how to experimentally distinguish beyond-LSWT effects from Stoner excitation effects is at present unclear. Lastly, the discrepancies between theory and experiment in Fig. 1b can also be attributed to limitations of the employed density functional theory calculations. The magnetic exchange interactions are computed from DFT and not fitted to experiment, so some differences are to be expected. The results based on DFT in Fig. 1(b) show that there is also a strong dependence on potential self-doping due to off-stoichiometry of the sample, which could lead to a small shift of the Fermi level.

Nevertheless, in our work the main role of LSWT is to establish the connection between the altermagnetic energy splitting and the chirality of the magnon bands, further linking to the theory of how magnons are detected in RIXS. This aspect should not be qualitatively changed by considering either beyond-LSWT or Stoner excitation effects.

To explicitly acknowledge these points and cite the recent literature, we have added the following to the manuscript, where Refs. [17–19] are the ones suggested by the reviewer:

Their energy spectrum and polarization offer unique insights into the microscopic mechanisms that distinguish altermagnets from conventional AFMs, whose magnonic properties are well understood [15]. The theoretical understanding of altermagnetic magnons is still being developed, with novel behavior due to anisotropies [16], magnon-magnon interactions [17–19] and altermagnetic Stoner excitations [20–22].

Regarding formatting, the manuscript lists some references with full names of authors (such as [18] Libor Smejkal, Alberto Marmodoro, Kyo-Hoon Ahn ...), which is non-standard practice, then reverts to standard practice in other cases (such as [50] K. Wildberger, P. Lang, R. Zeller, and P. H. Dederichs, ...).

We acknowledge the inconsistencies in the reference formatting and have revised this throughout.

Response to the reviewers

We thank the reviewers for their constructive feedback. Below, we address each comment point by point. The reviewers' reports are reproduced in black, our responses are in blue, and all changes to the manuscript and the Supplementary Information are highlighted in red.

Report of Reviewer #1

The authors have diligently addressed all comments from the reviewers and have revised the manuscript accordingly.

While the proposed explanation—that the MCD arises from a difference in magnon lifetimes—is plausible and self-consistent, it is, as Reviewer #2 also pointed out, a rather unconventional interpretation. My primary concern with the theoretical interpretation—specifically its lack of quantitative support—remains unchanged. The quality of the experimental data itself is not in question. The issue lies with the conclusive tone used to present the interpretation, which I believe is too strong given the nature of the evidence.

This interpretation is presented too conclusively in several key places. Specifically, I find the following phrases to be overstated:

In the Conclusion, the statement that your work “...completes the experimental demonstration...”. In the Introduction and Discussion, the use of “...unambiguously demonstrate...”. In the Discussion, the claim to have “...demonstrated a direct connection...”. In the Abstract, the phrase “...directly confirms the theoretical predictions...”.

The claim of having “completes the experimental demonstration” is particularly premature. A ‘complete’ demonstration would arguably require resolving the split peaks themselves, which should be achievable with an optimal combination of sample quality and a state-of-the-art RIXS instrument capable of sub-20 meV resolution at the Cr L-edge. Without such direct observation, the other strong phrases like “unambiguously demonstrate” and “direct connection” are similarly not yet fully justified.

Therefore, I would be prepared to recommend acceptance of this manuscript on the condition that these and similar interpretive claims are softened. The focus should be placed on the high-quality experimental observation of MCD, which is an excellent result in itself, while presenting the origin as a plausible but not yet definitive explanation.

Replacing these strong claims with more cautious phrasing—such as “our results suggest...,” “we propose a scenario where...,” or “the data is consistent with...”—would make the manuscript more robust and accurately reflect the current state of evidence. I hope these specific comments are helpful for further improving this important work.

We understand the Reviewer's concerns and in that light have decided to make the following changes.

In the Abstract:

“Our findings demonstrate a pronounced circular dichroism (CD) in the magnon peaks, with an azimuthal dependence that is consistent with the theoretical predictions and the g -wave symmetry.”

In the Introduction:

“Here, we introduce a novel and straightforward measurement protocol to demonstrate magnonic altermagnetism.”

In the Discussion:

“In this work, we established a connection between the circular dichroism of the magnon peak detected in RIXS and the predicted altermagnetic properties of the two circularly-polarized magnons in the *g*-wave altermagnet CrSb.”

In the Conclusion:

Here we retain our original formulation. This statement is not about establishing that CrSb is an altermagnet—that status is already firmly supported by several ARPES works that resolve the electronic *g*-wave altermagnetic splitting and symmetry in CrSb. The splitting of the magnon energy bands was already measured in MnTe, and in our work we complete the experimental demonstration of the predicted altermagnetic magnon properties by detecting circular dichroism at the magnon peak using RIXS, for which the only plausible origin is the magnon chirality.

“To conclude, our discovery of circular dichroism at the magnon peak completes the experimental demonstration that altermagnetic magnons indeed possess all their predicted properties...”

Report of Reviewer #2

The authors proposed a good idea that bypasses the problem raised by the reviewer #1 and me. I’m very pleased to recommend publication of the revised article in Nature Communications.

We thank the reviewer for the positive assessment and recommendation.

Report of Reviewer #3

The authors have resolved all technical questions of 3 referees, and correspondingly improved the manuscript. There is no doubt that their effort to introduce novel techniques for probing chiral magnons (an emerging topic on its own, of potential application significance as well for magnonic technologies) in altermagnets could have a large and broad impact. This has been recognized already, as in the viewpoint, <https://www.condmatjclub.org/2025/06/>, written for the highly selective Journal Club for Condensed Matter Physics. So, I recommend the paper be published as is.

We thank the reviewer for the supportive evaluation.

In this paper, Biniskos et al. carry out careful resonant inelastic x-ray scattering measurements and report unambiguous circular dichroism in magnon intensity in the altermagnet CrSb by particularly performing wide azimuthal scans for two opposite altermagnetic domains. This subject is very topical in condensed matter physics and hence is intensively studied.

Although, as will be mentioned, I think that the theoretical treatment is insufficient, the dichroic features are exhibited in magnon intensity without ambiguity. From an experimental point of view, this is an important progress, which merits publication in Nature Communications.

I think that several revisions are necessary. Please carefully consider following comments and correctly reflect them in the revised manuscript.

(1) The authors should distinguish “splitting” from “dichroism in intensity”.

line 116-117: **This can be interpreted by comparing with the theoretically determined magnon bands**

line 126-127: **Fig. 2(f) leads to the predicted azimuthal dependence $I_{CD}(\phi) \propto \cos(3\phi)$,**

Fig. 2(f) indicates the azimuthal dependence of energy splitting in magnon bands. Hence it does not (simply) explain the observed dichroism in intensity and its ϕ dependence.

line 190-191: **altermagnetic magnons indeed possess all their predicted properties – energy splitting and circular polarization –,**

The authors observed the circular dichroism in magnon intensity, but the energy splitting in magnon bands was not detected in this paper.

(2) If my understanding is correct, $R_{CD}(\omega)$ in Eq. S53 should be a dark green line indicated in Fig. R1. However, instead of this negative-positive (or positive-negative) spectrum, a single peak is observed in dichroic intensity. Therefore, the authors should describe in the main text that the theoretical calculation presented in the Supplementary Information does not interpret the experimental observation and more sophisticated theoretical approach is necessary. Accordingly,

line172-174: **This explains the correlation between the azimuthal dependence of the experimental circular dichroism (Fig. 3(b)) and the azimuthal dependence of the two magnon bands and their energy splitting (Fig. 2(f)).**

This is not correct.

Abstract line 11: **the theoretical predictions and**

This is not correct.

Figure R1: Expected dichroic spectrum (dark green) for split peaks with opposite chirality (red and blue). Lorentzians with $\Gamma = 33$ meV. Energy splitting is 22 meV.